# Emotional Labor and Burnout among Teachers: A Systematic Review

**DOI:** 10.3390/ijerph182312760

**Published:** 2021-12-03

**Authors:** Anna Kariou, Panagiota Koutsimani, Anthony Montgomery, Olga Lainidi

**Affiliations:** Department of Educational & Social Policy, School of Social Sciences, Humanities and Arts, University of Macedonia, 54636 Thessaloniki, Greece; anna.kariou@uom.edu.gr (A.K.); koutsimani@uom.edu.gr (P.K.); lndolga@gmail.com (O.L.)

**Keywords:** burnout, emotional labor, education, teachers, systematic review

## Abstract

A significant amount of emotional labor takes place during teaching. Teaching is a multitasking profession that consists of both cognitive and emotional components, with teachers engaging in emotional labor on a daily basis as an instrumental part of achieving teaching goals and positive learning outcomes. The purpose of the present review was to explore the relationship between emotional labor and burnout in school settings. The review focused specifically on teachers from elementary and high schools, between January 2006 and August 2021, and 21 studies fit the inclusion criteria. Overall, the review of the literature supports the significant associations between burnout and emotional labor with the majority of results pointing to the consistent relationship between surface acting and burnout. However, the results regarding the association of deep acting and naturally felt emotions with burnout were mixed. There is considerable scope for improvement in our study of emotional labor in terms of the study designs we employ, the variables we study and our appreciation of the historical and cultural factors that moderate and mediate the relationship between emotional labor and burnout.

## 1. Introduction

Emotional labor has always been a characteristic of teaching, but it is set to become even more important in the 21st century. The rapid expansion of the service economy globally has resulted in shifting normative expectations for service employees in all industries [1]. Increasingly, workers are expected to engage in more emotional labor [2] in all sectors, which translates into the need for employees to express organizationally desired emotions during interpersonal interactions [3]. In this sense, the feelings and behaviors that are used in the workplace become exchanged commodities with economic value attached to them [2]. This is consistent with the literature indicating that employees can experience burnout as a result of the congruence and discordance between their personal emotional states and occupational expectations [4]. The aforementioned is particularly relevant for the profession of teaching, where student and parent expectations have changed significantly over the last 30 years.

The phenomenon of institutional isomorphism in higher education is having an impact on education at all levels. Institutional isomorphism refers to the increasing practice of using national rankings, and how this influences efforts by universities and colleges to structure programmes to remain competitive with peer institutions, rather than being driven by the historical and/or local values of an institution [5]. The trickledown effect for teachers in elementary and high school education is a greater focus on exam results, which results in more emotional labor to keep the balance between affective neutrality and pastoral interest in students’ wellbeing [6]. In short, there is an increasing movement towards students as customers which ramps up the need for teachers to be constantly under emotional management.

### 1.1. The Modern Phenomenon of Emotional Labor

Emotional labor (EL) is the effort, planning and control needed to express organizationally desired emotions during interpersonal transactions [3]. Thus, it is a stressor factor but a necessary one in order for employees to regulate their feelings and expressions to achieve organizational goals [7]. Keller et al. [8] notes that emotions can influence teachers’ behaviors, students’ behavior and outcomes, and teachers themselves have mentioned that they use emotional labor on a regular basis. While emotional labor researchers have focused on service-related professions, relatively little attention has been given to educational settings where the need for daily interactions is a defining characteristic of work, in terms of unique, stable, repeated, intense and long-term daily interaction [9].

A significant amount of emotional labor takes place during teaching [8]. Teaching is a multitasking profession that consists of both cognitive and emotional components, such as teaching and designing the curriculum but also expressing or hiding true emotions or expressing the appropriate emotion for the situation, even if it is not true [10]. Teaching can be like acting in a live play or stand-up comedy [11]. Teachers are constantly exposed to the criticism of their students, parents, coworkers and principals, and are forced to deal with numerous emotional situations and at the same time be a role model for their students [12,13]. Therefore, unlike other employees in the service sector, teachers engage in emotional labor not just to align with the prescribed emotional-display rules, but also because they see such efforts as instrumental in reaching their teaching goals and positive learning outcomes [14]. It is a complex combination of decision making and emotional regulation [10]. For example, they need to manage their own anger in the classroom, show sympathy for awkward situations, show care for the progress of their students, continuously encourage them and their parents, and collaborate with their colleagues. Finally, problematic behaviors are more frequent in schools compared with other work environments [15].

### 1.2. The Components of Emotion Labor

In the following section, we will review the three core concepts within EL; surface acting (SA), deep acting (DA) and naturally felt emotions (NFE). Noor et al. [16] describes SA as changing one’s outward emotional expressions without attempting to feel the emotions displayed and deep acting as changing one’s outward emotional expressions, and at the same time, attempting to feel the emotions displayed. Thus, DA is the effort to truly feel the appropriate emotion according to the situation [17], while SA is the hiding or down regulating of felt emotions and faking false feelings [18]. Naturally felt emotions are defined as the effort involved for a person to express the feelings that they are actually experiencing in a genuine way [10].

Historically, the concept of EL evolved from the work of Hochschild [2], who originally talked about two main elements; surface and deep acting. More recently, Diefendorff [19] (2005) introduced the element of naturally felt emotion. Surface and deep acting are considered by Hochschild [2] and Grandey [7] as the appropriate response that arises from the expectations of the job. According to Yilmaz et al. [17], the fundamental difference between the dimensions (i.e., SA, DA, and NFE) is the level of internalization of behaviors. Surface acting involves non-internalized emotions, while naturally felt emotions involve internalized emotions. The internalization level in deep acting is more than that of surface acting, but less than naturally felt emotions. Surface acting can lead to feelings of inauthenticity or emotional dissonance, thus discouraging employees from reciprocating in the form of positive attitudes and behaviors [20]. Surface acting strategies that result in dissonance and exhaustion mean lower levels of satisfaction of individuals’ basic psychological needs at work, which are themselves known to predict impaired work functioning [21]. Conversely, there is evidence that deep acting does not require as many cognitive resources as surface acting [22].

### 1.3. Burnout and Emotional Labor

Burnout is a psychological syndrome that involves a prolonged response to chronic emotional and interpersonal stressors on the job [23]. The three key dimensions of job burnout are exhaustion, feelings of cynicism/depersonalization, and a sense of professional inefficacy/lack of accomplishment [24]. According to Montgomery and Maslach [25]; exhaustion is the individual stress dimension of burnout, and it refers to feelings of being physically overextended and depleted of one’s emotional resources; cynicism (or depersonalization) refers to a negative, callous, or excessively detached response to other people; and inefficacy (or lack of accomplishment) refers to a decline in one’s feelings of competence and successful achievement in one’s work.

Burnout has been linked with emotional and cognitive loss of control [26], as employees who suffer from burnout often experience difficulties in regulating their behavior, possibly due to executive control deficits [27,28]. Diminished executive control abilities could mediate the relationship between burnout and emotional labor. That is, burned-out teachers might be less able to apply strategies for regulating their emotions as a result of executive control deficits.

Educators are susceptible to the impact of emotional labor on burnout [29,30]. Teaching is a highly demanding profession, which requires both excellent knowledge of the subject and at the same time requires unlimited psychological resources for coping with everyday emotional challenges. The conservation of resources model (COR) is often mentioned as an attempt to better understand the source of burnout and the mediating role of emotional labor [31]. In essence, individuals want to maintain objects, skills and resources valuable to them, and when they sense a threat or when resources are used, burnout increases. The COR model has been used to explain the significant reduction in healthcare employees’ psychological capital levels during mandatory confinement due to COVID-19 [32], which results in resource depletion.

In the case of teachers’ emotional labor, loss of resources is one of the main drivers of burnout [10]. The COR model shows that these resources are not unlimited, and that is the time when problems arise. Poor miscalculation of emotional labor dynamics and failure to use strategies can lead to burnout. The resources model explains the imbalance between emotional demands and the resources available to regulate emotions. Teachers are guiding, teaching, collaborating, behaving in formal and informal ways and at the same time less likely to be reflecting on their personal problems [17].

Burnout is a potential cause of teachers’ dropout and retirement [8] and emotional labor contributes to these outcomes. Teaching is demanding and exhausting, and this can drive some teachers throughout their career to try to ameliorate the impact of their work by moving to ‘less demanding’ positions within the educational system such as administration, library services and management. In terms of emotional labor, NFE can help to conserve their resources [10], in contrast to surface acting which consumes resources as it involves a greater investment of resources than deep acting [33]. Congruently, Lenine and Naring [30] pointed out that burnout in teaching can follow an exponential growth pattern as teachers are impacted by emotional labor practices. In a longitudinal study, the results of Philipp et al. [33] highlight how the COR model can work in practice; as time passes, emotionally exhausted educators use more surface acting in order to reduce the use of their resources.

Theories of emotional inhibition and emotional repression have been associated with burnout. Montgomery et al. [34] suggested that the inhibition of emotions is associated with increased physiological chronic arousal, which is easily connected to negative health outcomes. Additionally, Yilmaz et al. [17] pointed out that few studies have examined emotional labor in relation to personality characteristics (with some exceptions [35,36]).

The relationship between EL and burnout is a mixed picture. SA is a positive predictor of burnout [37,38,39,40] while DA on the contrary may even result in positive psychological well-being [7,38]. Park et al. [10] suggests that naturally felt emotions can help conserve resources, in accordance with the COR model, and that interpersonal influence and deep acting reduce the effect of surface acting in burnout. According to Näring, Briët, and Brouwers [30], surface acting leads to higher levels of burnout. Naturally felt emotions were negatively related to burnout and positively related to job satisfaction [29]. Yilmaz et al. [17] reported that surface acting and naturally felt emotions are associated with the emotional exhaustion and depersonalization of teachers, while Akin et al. [11] reports that deep acting decreases burnout while surface acting increases the effect. Wrobel et al. [9] summarize the beneficial points of deep acting. Deep acting is positively associated with job satisfaction and personal accomplishment. Deep acting techniques require less energy and resources so are less emotionally exhausting.

To the best of our knowledge, a comprehensive review of the literature, including a summary and analysis of existing studies on this issue, has not been conducted to date. In order to fill this gap, the purpose of the present study was to answer the following questions: (1) What is the relationship between surface acting and burnout? (2) What is the relationship between deep acting and burnout? (3) What is the relationship between naturally felt emotions and burnout? (4) What are the remaining gaps in the literature with regard to the study of emotional labor and burnout among teachers from elementary and high schools?

## 2. Materials and Methods

### 2.1. Objectives

The aim of the present systematic review was to explore the extent of the relationship between burnout and emotional labor, which has been empirically studied among educators. The review will focus specifically on teachers from elementary and high schools with the purpose of increasing the understanding of the relevant literature and to inform future studies.

### 2.2. Search Methods

A comprehensive database search was conducted in order to identify the studies that examined the relationship between burnout and emotional labor among elementary and high school educational personnel. The steps for the identification of all relevant studies were informed by the Preferred Reporting Items for Systematic Reviews and Meta-analysis (PRISMA) statement (Page et al. [41]; see Figure 1). To operationalize the research questions and search strategy, we were guided by the PICOS approach [42]. The PICOS acronym, which stands for population, intervention, comparators, outcomes and study design, was used to refine the research questions and develop the terms used in searching the literature.

Specifically, the online databases of PubMed, SCOPUS and Web of Science were searched for studies published between January 2006 and August 2021 using “burnout” and “emotional labor” as search terms. Initially, “emotional labor” and “burnout” were the search terms used in every database. After excluding duplicates and non-English papers, the following search terms were included in the search process: teacher, schools, emotional exhaustion, elementary school, middle school, high school, primary school, and educator.

Moreover, in order to account for any studies that did not turn up in the database search, we proceeded to manually scope the cited articles of all eligible studies.

The eligibility criteria included peer-reviewed, full-text studies written in the English language which utilized quantitative designs (cross-sectional and longitudinal) and measured burnout and emotional labor by using reliable research tools. In addition to these criteria, we considered only studies that examined primary and high school educational personnel (i.e., teachers, teaching assistants and principles, and other types of educational staff) who worked either in typical or special education in both public and private schools. Initially, we screened the abstracts of the articles in order to determine their relevance to the systematic review. If an abstract fit the eligibility criteria, then we proceeded to a full article appraisal.

### 2.3. Results

The database search yielded 3099 records which resulted in 17 that fitted the eligibility criteria. Manual scoping of these studies resulted in four additional articles. In total, 21 studies were included in the systematic review. Study characteristics are reported in Table 1. Specifically, in 2006, 2008, and 2010, one study was published for each year followed by three studies in 2011, two in 2013, three in 2014, two in 2015, one in 2016 and two in 2017. Three studies were published in 2019 and finally two studies were published in 2021. With respect to the studies’ sample characteristics, the overall sample size was 7193 (57.8% women; 42.2% men) including teachers, administrators, and other types of educational personnel. Concerning the school levels, ten studies examined the educational personnel of elementary schools, two studies focused on the examination of high school educational personnel and seven studies assessed both elementary and high school educational personnel. One study [18] did not report the school level.

Regarding the research tools, most studies measured burnout using the Maslach Burnout Inventory (MBI) [23]. The majority of the studies assessed emotional labor with the Emotional Labor Scale (ELS) [38]; four studies measured emotional labor with the Teacher Emotional Labor Strategy Scale (TELSS) [43], one study with the Emotion Work Scale by Zapfetal et al. [44] and one study with the Emotional Labor Scale [45]. Lastly, concerning the design of the studies, 18 of them utilized a cross-sectional design and three studies were of a longitudinal design. Concerning the geographical distribution of the studies, the papers reported research from the following countries: Israel, Poland, India, Netherlands, France, Australia, the UK, Pakistan, Nigeria and Malaysia. Two studies came from Germany. Finally, three studies were reported from the following regions: China, Turkey and the USA.

## 3. Summary of Studies

Overall, the review of the literature supports the significant associations between burnout and emotional labor, with the majority of results pointing to the consistent relationship between surface acting and burnout. However, the results regarding the association of deep acting with burnout were mixed. Specifically, Naring et al. [30] found that surface acting predicted exhaustion and cynicism while the total emotional labor score was a predictor of personal efficacy. Philipp et al. [33] provided some interesting insights regarding the nature of the relationship between burnout and emotional labor. Particularly, what the researchers observed was that although deep acting negatively affected exhaustion feelings, when the teachers started to feel emotionally exhausted, they began employing surface acting. Cheung et al. [29] found that higher surface acting scores predicted higher exhaustion and cynicism levels while deep acting led to a reduction in reduced personal efficacy. Moreover, the researchers also found that naturally felt emotions were negatively related to the total burnout score.

Additional research findings show the effects of surface acting in higher exhaustion [14,16,18,46]. Further studies corroborate this observation, but they also show the role of deep and natural acting as mitigators of exhaustion [11,35], possibly suggesting that teachers who invest in being empathetic towards their students, their students’ parents and their co-workers might gain more gratification from their job and thus, experience less exhaustion. Wrobel et al. [9] observed that deep acting may be motivated by a high level of empathy, and this notion could be supported by the results of Park et al. [10] who found that deep acting was negatively related to reduced personal efficacy, indicating that genuine efforts towards a compassionate path can lead to a higher sense of one’s beliefs regarding their job attainments. In addition to this finding, the researchers observed that surface acting was positively correlated with all three burnout dimensions. Likewise, Anomneze et al. [47] found that surface acting predicted higher exhaustion levels while deep acting predicted lower exhaustion and also higher cynicism levels. Other studies show that both surface and deep acting are positively related to exhaustion [9] while others give no support on the role of deep acting in the burnout experience [17]; the latter researchers also found that surface acting predicted both exhaustion and cynicism.

More recent studies provide further insights regarding the association between emotional labor and burnout. Particularly, Lee et al. [48] found that surface acting was positively associated with the total burnout score while naturally felt emotions were inversely associated with the total burnout score. Moreover, deep acting was negatively related to reduced personal efficacy [48]. Likewise, other scholars observed that surface acting led to an increase on all three burnout dimensions [49] and on the total burnout score [20]. Interestingly, Tsang et al. [49] further found that deep acting resulted in a decrease in cynicism and reduced personal efficacy. Thus, the results of Lee et al. [48] and Tsang et al. [49] provide additional support to the view which holds that deep acting can mitigate burnout. In addition to these studies, the study of Fouquereau et al. [50] provided important insights on how surface and deep acting might interact with each other in relation to burnout onset. Specifically, the researchers observed that deep acting was positively associated with emotional exhaustion, suggesting that surface acting could eliminate the positive effects of deep acting.

Concerning the studies that examined burnout as a predictor of emotional labor, they indicate that greater exhaustion scores predict emotional labor [8]. Correspondingly, further studies also show that higher burnout levels are related to higher surface acting while lower burnout levels are related to NFE; no associations were observed between burnout and deep acting [15,36].

The associations between emotional labor and burnout are well-established. However, underlying mechanisms could affect the strength of the relationship. Park et al. [10] found that the strength of the relationship between surface acting and reduced personal accomplishment was moderated by interpersonal influence, the ability of one individual to affect with flexibility the behavior or beliefs of others [51]. Park et al. [10] also showed that surface acting and naturally felt emotions were related to reduced burnout and increased levels of organizational citizenship behavior at the interpersonal level. Overall, seven studies [10,17,29,30,48,49,52] showed that personal accomplishment was strongly related to emotional labor and burnout.

Basim et al. [35] examined the mediating role of emotional labor between personality and emotional exhaustion. Neuroticism and extraversion significantly predicted emotional exhaustion, while neuroticism significantly predicted surface acting, openness to experience significantly predicted deep acting and agreeableness significantly predicted naturally felt emotions. Emotional labor had a partial mediating role in the relationship between personality and emotional exhaustion. Khalil et al. [36] examined emotional labor as a mediator between personality and burnout, but unfortunately the authors failed to distinguish between the different traits of the Big Five personality traits in their paper, which makes it difficult to reach any definitive conclusions.

Silbaugh et al. [13] showed that emotional labor could be a mediator between emotional intelligence and burnout through the use of surface acting. Cheung et al. [29] focused on the moderator role of psychological capital (PsyCap). PsyCap was negatively related to burnout; PsyCap was positively related to natural felt emotions; PsyCap buffered the negative association of surface acting on depersonalization.

Gender is also an important individual factor that could affect both the emotional labor and burnout experience. Akin et al. [11] found that women showed higher emotional labor through the use of deep acting. Yilmaz et al. [17] found that men were using more surface acting, but that burnout levels did not differ between men and women. Congruently, Anomneze et al. [47] found that gender per se was not significantly associated with emotional labor but that the marital status was significant, as single people used more deep acting strategies. Noor et al. [16] showed that work–family conflict mediated the relationship between emotional labor and burnout.

Organizational factors can have a key role in the association between emotional labor and burnout. Lee et al. [48] showed that teachers’ burnout was positively associated with teachers’ intent to leave their jobs but burnout mediated the relationship between surface acting and turnover intention. Yao et al. [46] found that the school climate negatively affected surface acting but positively affected deep acting. Tsang et al. [49] found that burnout was positively related to market culture, which increased emotional exhaustion and depersonalization by raising surface acting. On the other hand, burnout was negatively related to hierarchy culture which decreased emotional exhaustion and depersonalization by reducing surface acting and increasing deep acting. Anomneze et al. [47] showed that deep acting and perceived organizational support were negatively related to emotional exhaustion. Cheung et al. [29] observed that the expression of naturally felt emotion was positively related to job satisfaction. Naring et al. [30] found that emotional exhaustion and depersonalization associated with quantitative demands, less control, and less social support and were related to more surface acting. Yilmaz et al. [17], found that school type was also an important factor. Elementary school teachers used more surface acting and deep acting in comparison to high school teachers, who used natural felt emotions. Depersonalization was higher in high school teachers, and Akin et al. [11] and Lee et al. [48] showed that teachers used more natural felt emotions. Yilmaz et al. [17], showed that teachers used more natural felt emotions but administrators used more surface acting. Maxwell et al. [18] observed that principals used more surface acting than deep acting.

## 4. Discussion

The aim of the present review was to explore the relationship between emotional labor and burnout among teachers. All emotional behaviors influence burnout but the majority of the studies focused on the role of surface acting. The results of the present review revealed the consistent association between surface acting and emotional exhaustion [8,9,10,11,16,29,30,33,35,46,47,49]. Nevertheless, whether burnout is the result of emotional labor or burned-out employees start to employ surface acting responses as they lack the resources to apply more genuine and empathetic strategies (i.e., deep and natural acting) is not yet well understood. Few studies have examined burnout as the cause of emotional labor [8,15,36], and those that did have shown that burnout leads to greater surface acting strategies and emotional labor. This association is consistent among the studies, suggesting that when emotional exhaustion and fatigue rises, surface acting rises too [16,33,50,52]. Can we assume that burnout is the cause and surface acting is the effect? It seems that when our organizational circumstances present teachers with increasingly difficult work conditions, teachers increasingly employ surface acting to deal with it, which can be become a reinforcing cycle—leading to greater feelings of burnout.

However, emotional labor has an interesting bright side. We should not forget the fact that teaching can also be a source of positive emotions. Naturally felt emotions and deep acting play the role of a moderator and they are able to buffer the negative effects of burnout. Both naturally felt emotions and deep acting can ameliorate the impact of burnout [10,11,35,48,49]. This is consistent with the notion that it is not the performance of emotional management that is harmful, but rather it is the expectation of having to manage emotions that is harmful in the workplace [53].

Emotional labor could be an indicator that reveals teachers’ performance [17], a hidden window to their level of professionalism. Cheung et al. [29] noticed that through deep acting teachers gain job satisfaction and better job performance. Khalil et al. [36] and Basim et al. [35] found limited support on the relationship between personality traits and emotional labor, but there is not yet enough evidence to warrant serious consideration of personality traits as a component of emotional labor. Job position seems to have an impact. Maxwell et al. [18] found that principals more often hide their emotions, which predicts a high level of burnout and job satisfaction. Surface acting was the main strategy that they used, maybe as a result of their higher burnout level or the fact that principals tend to be men. Thus, school principals have a much higher level of burnout compared to the other school staff which results in poorer psychological health. Such findings are worrying given that the sustainability of the education system relies heavily on the wellbeing of line managers.

Emotional labor is influenced by many factors. Structural variables such as gender, marital status, years of experience, school type, job description, individual and organizational factors have a role to play in the development and maintenance of emotional labor. School type appears to have significance also. Akin et al. [11] showed that deep acting and natural felt emotions were more commonly used by teachers in private schools. Therefore, school type is another focus area that has been underestimated, and the different types of schools (i.e., private, public, university, special, urban, musical, athletics, ecclesiastic, etc.) may differ in the type and level of emotional labor and burnout. Moreover, the size of the school, funding source, and infrastructure may have an impact on burnout and inevitably on emotional labor. In order to fully understand these relationships, future research should include more variables related to school type and school governance structures.

### 4.1. Gaps in the Literature

The review provides us with a useful picture of what areas are in need of more investigation in emotional labor. In particular, the following areas are worthy of more discussion and investigation; theoretical development, the role of students in the emotional labor process, culture, gender, and whether burnout and emotional labor happen in parallel.

All the studies reviewed cover the relevant EL literature in their introduction sections. However, the coverage and discussion of the theoretical background of emotional labor is inadequate. The majority of studies do not explicitly discuss theoretical issues, and the ones that do tend to link emotional labor to either the Conservation of Resources [31] model and/or the Job-Demands Resource Model [54]. The aforementioned theories provide a useful backdrop to the drivers of burnout, but the theoretical development of the field would benefit from other approaches that are more focused on emotional management. For example, Pekrun’s [55] control-value theory of achievement emotions can provide an interesting way to link teacher and students’ emotional regulation. In this theory, Pekrun [55] suggests that subjective control and value appraisals are central antecedents of both teachers and students’ emotional experiences in the learning and achievement context, and that both actors might experience negative emotions such as anxiety, anger, or frustration, if a classroom situation is inconsistent with their goals. Equally, self-determination theory (SDT) [56] is interested in the functioning of the self, that is, its organization of experience, and its regulation and integration of impulses, emotions, motives, and values. According to SDT, we have propensities toward autonomy, competence, and relatedness and the flourishing associated with them, which require specific social nourishments and supports. SDT provides us with a way to view how internal and external motivations of teachers shape their emotional regulation. Both control-value theory and SDT are just two examples of alternative approaches to emotional labor, but the broader point is that there is a challenge for researchers in terms of how they utilize different theoretical approaches to understand the drivers and mechanisms of emotional labor.

The majority of papers in the review investigate emotional processes as they occur in a given teaching situation, but we have relatively little information as to what causes teachers’ emotional reactions in class (with the exception of the Keller et al. [8] paper). For example, research has shown the crucial importance of students’ (mis)behavior in the context of teacher EL and exhaustion [57,58]. Thus, the impact of emotional labor on pupils’ progress and achievement might also be a fruitful topic for future study.

Cultural context was not examined in great depth in any of the papers. Historically, emotional labor as a concept was developed from studies of teachers’ emotional labor and emotions from Western samples in general educational contexts. However, culture plays a fundamental role in shaping individuals’ cognitive, emotional, and motivational processes [59]. In terms of differences, research has suggested that individuals in Eastern countries tend to display lower life satisfaction, less positive emotions, and more negative emotions compared to individuals in Western countries (e.g., [60,61]).

Gender differences were reported in the majority of studies, but beyond this there was very little substantive discussion of gender. Research suggests that women do more emotional work than men [2]. The reasons may vary; women choose jobs that require more emotions (nursing, teaching, etc.) and at the same time have to complete more tasks in everyday life due to an unequal distribution of non-work roles and demands (i.e., wife, mother, maintenance requirements, household responsibilities). Recent studies have shown that women tend to be more vulnerable to fatigue as a result of overloaded responsibilities and everyday conflicts [16], and there is some evidence that women experience emotional exhaustion more than men, while men experience depersonalization more than women [62]. Moreover, Akim et al. [11] found women teachers use deep and surface acting strategies more often than men. There is a need for us to understand how expectations around gender may influence the experience of emotional labor for teachers.

The reviewed studies do not shed light on whether emotional labor is an independent variable, or whether burnout and emotional labor occur in parallel. We do not know exactly when somebody starts experiencing burnout and thus it is difficult to know whether burnout or emotional labor happens first. Teachers are reporting increased levels of burnout, while reported levels of emotional labor are also very high. Since burnout is consuming a large number of resources, teachers choose to engage in surface acting, as this is expected to be the less resource-consuming strategy (e.g., faking a smile is perceived to be less demanding than trying to induce the actual emotion that will lead to smiling). In contrary, teachers expect that deep acting or even the expression of naturally felt emotions will be more demanding in the present, as it might put them in a situation that will lead to further consumption of cognitive and emotional resources. Thus, when it comes to emotional regulation, it is probable that teachers fail to plan ahead and choose the option that appears less demanding in the present—especially when those resources are already compromised due to the experience of burnout. Future researchers need to explore whether emotional labor is an ongoing coping strategy to deal with burnout rather than a cause or antecedent of burnout.

### 4.2. Limitations

Overall, the reviewed papers rely on self-report measures, thus we are unable to assess the degree to which the measurement of emotional labor is impacted by social desirability and common method biases. In this vein, the impact of emotional labor would be more meaningful if the reviewed research linked it with more objective data (e.g., records of absenteeism, turnover, or performance). The majority of reviewed papers were cross-sectional studies and the field would benefit from more longitudinal designs and the use of more innovative approaches. Only one paper, Keller et al. [8], used an Experience Sampling Methodology (ESM) approach. ESM assessment allows for intra-individual analyses of teachers’ emotional processes, and investigating intra-individual functioning is a core goal in personality psychology [63], but the majority of research on teachers’ emotions focuses on inter-individual differences. Longitudinal and repeated measures approaches to data collection are particularly important when we consider that teachers develop their emotional regulation strategies through experience and we may find that emotional labor intensifies through time.

### 4.3. Practical Implications

Kinman et al. [52] suggested that enhancing teachers’ emotional skills can help to reduce the negative impact of emotional labor. Addressing specific variables (e.g., level of burnout and level of education) may lead to the improved management of emotions in teaching [15]. Lee et al. [48] recommended that teachers could be taught to use cognitive strategies of deep acting and genuine expressions. In order to reduce turnover intentions, Maxwell et al. [18] recommends training in interpersonal skills, training employees to work cooperatively, better administrator structures, management workshops, and mentoring practices from more experienced teachers in order for teachers to improve empathic concern skills. Diefendorff and Gosseran [64] noted that by informing teachers about emotional labor, and by training them in rules related to the demonstration of emotions and emotional labor, their levels of emotional labor and burnout could be moderated [15]. Philipp et al. [33] and Wrobel et al. [9] suggest training in deep acting techniques, and approaches that strengthen dedication to teaching. Yon et al. [46] pointed out the necessity not only to improve the individual but to make an effort to improve the school organizational environment which strongly predicts emotional labor and reduces burnout.

The aforementioned highlight that emotional management approaches for teachers are treated as continuous professional education rather than a core set of skills to be taught during initial teacher training. This begs the question as to whether current teacher training places too much emphasis on technical skills (e.g., curriculum development, assessment methods, teaching methods) and not enough on emotional regulation strategies and self-reflective practice. Prioritizing teacher well-being is especially important when we consider that the teacher–student relationship is unique, in that we typically have one person being responsible for a large group of children/adolescents over a prolonged period of time. Moreover, teachers are required to accept legal and ethical responsibilities (as well as parental expectations) for their students beyond delivering knowledge.

## 5. Conclusions

The results of the present systematic review are in broad agreement with the meta-analysis of Hόlsheger and Schewe [65] on emotional labor in service organizations generally, which found that surface acting displayed a positive relationship with emotional exhaustion, depersonalization, and psychological ill-health, and a negative relationship with job satisfaction and organizational attachment, whereas deep acting presented a positive association with emotional performance. Our review indicates that among teachers, SA is associated with increasing levels of burnout. Trends suggest that emotional labor is likely to become even more intense in the teaching profession as a result of higher expectations of service from both students and parents. Thus, it is quite likely that the future picture is one where teachers who experience more emotional labor are more likely to be more emotionally exhausted and less satisfied with their work, and also more likely to depersonalize their students. This has the potential to be a vicious circle. Finally, there is considerable scope for improvement in our study of emotional labor in terms of the study designs we employ, the variables we study and our appreciation of the historical and cultural factors that moderate and mediate the relationship between emotional labor and burnout.

## Figures and Tables

**Figure 1 ijerph-18-12760-f001:**
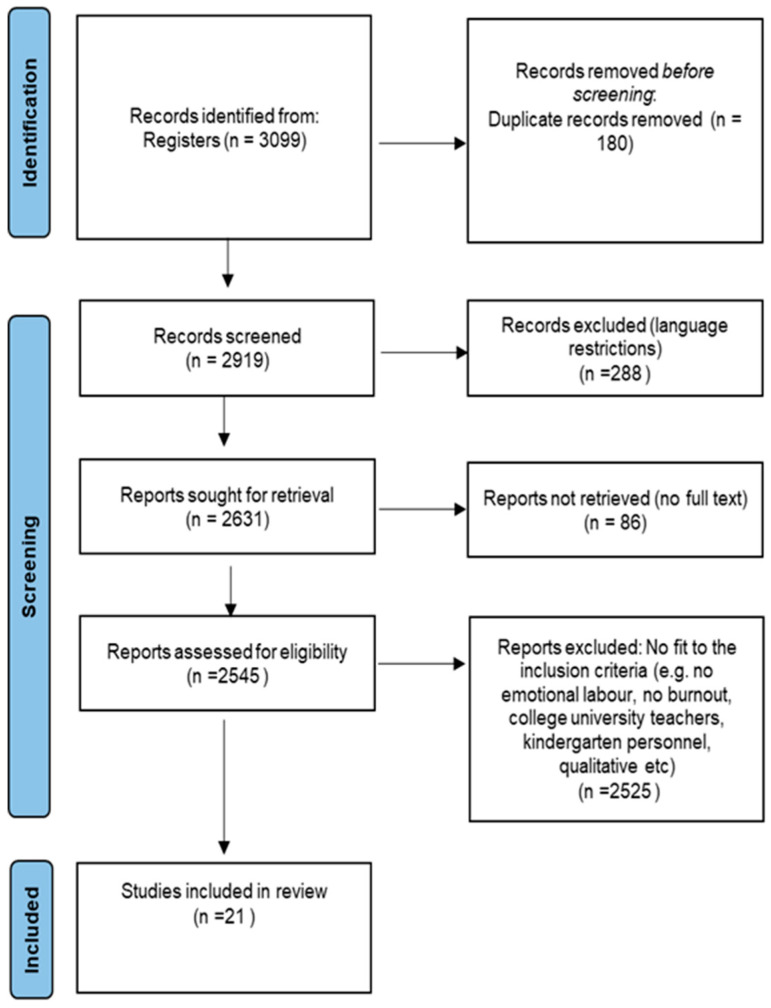
Identification of studies examining burnout and emotional labor among primary and high school educational personnel.

**Table 1 ijerph-18-12760-t001:** Description of studies in the systematic review.

No	Author(s) and Year	Sample % Women	School Type	Country	Burnout Measure	EL Measure	Main Findings/Outcomes
1	Akin et al. (2014)	68.6%	ELE	Turkey	MBI	Emotional Labor Scale	Turkish primary school teachers mostly engage in GE in their relationships with students. Female teachers useDA and SA strategies more often than males. EL is a significant predictor of TB
2	Anomneze et al. (2016)	54.7%	ELE	Nigeria	MBI	Teacher Emotional Labor Strategy Scale adapted by Yin	Only DA and perceived organizational support significantly predicted DP
3	Basim et al. (2013)	53%	ELE & HS	Turkey	MBI	Emotional Labor Scale	Personality traits (neuroticism and extraversion) were found to predict EL. All dimensions of EL were found to affect EE. SA was the only mediator in the relation between neuroticism and EE
4	Cheung et al. (2011)	64.4%	ELE	China	MBI	Emotional Labor Scale	SA correlated more positively with TB while DA was only negatively related to lack of PA. PsyCap moderated the association between EL and TB
5	Fouquereau et al. (2019)	-	-	France	MBI—General Survey	Emotional Labor Scale	High levels of all labor strategies but DA much higher. High level of DA leads to less EE.
6	Keller et al. (2014)	51.2%	ELE	German	MBI	Frankfurt Emotion Work Scale	Teachers reported suppressing or faking emotions during a third of all lessons. Enjoyment, anxiety and anger on an intra individual level predict EL. On inter individual level anger evokes EL.
7	Khalil et al. (2017)	-	ELE	Pakistan	MBI	Emotional Labor Scale	All dimensions of EL were found to affect TB. Negative relationship between personality and TB. Positive relationship between personality and EL.
8	Kinman et al. (2011)	72.6%	ELE	UK	MBI	Frankfurt Emotion Work Scale	Positive relationships between EL and PA. Experienced teachers reported higher levels of EL.
9	Lee et al. (2019)	41.5	ELE	USA	MBI-ES	Teacher Emotional Labor Strategy Scale adapted by Yin	Teacher TB was positively associated with SA and negatively associated with GE. Teacher TB was positively associated with turnover intention. Use of DA leads to a feeling of PA. Teacher TB mediated the relationship between SA and turnover intention.
10	Maxwell et al. (2017)	51.8%	-	Australia	MBI	Emotional Labor Scale	SA—Hiding emotions had an inverse relationship with ΤB, wellbeing and job satisfaction. DA demonstrated no significant associations with outcome variables
11	Naring et al. (2006)	26.3%	ELE	Netherlands	MBI-NL-Ed	Emotional Labor Scale Dutch Version	SA and suppression are significantly related to DP. EL related to PA. SA related to EE
12	Noor et al. (2011)	100%	ELE	Malaysia	Teacher Burnout Questionnaire	Emotional Labor Scale	SA was positively associated with SA and DP
13	Park et al. (2014)	80%	ELE and HS	USA	MBI	Emotional Labor Scale	GE were positively related to OCB—I. Interpersonal influence could act as a resource, buffering the negative effects of SA on reduced PA. Interpersonal influence training may reduce the negative effect of SA. DA strategy might be beneficial over SA
14	Philipp et al. (2010)	80%	ELE	German	MBI	Emotional Labor Scale	Use of DA leads to less EE. DA could be health beneficial
15	Richardson et al. (2008)	48.7%	HS	USA	MBI	Emotional Labor scale	Significant and positive association between EE and intent to leave the job. Significant and positive link among emotive dissonance and intent to leave
16	Silbaugh et al. (2021)	40.5%	ELE and HS	India	MBI—General Survey	Emotional Labor Scale	SA is determined to be a partial mediator between EI and TB. Significant negative relationship between EI and SA. Significant positive relationship between SA and TB. Significant negative relationship EI and TB.
17	Tsang et al. (2021)	84.4%	ELE	China	MBI-ES	Teacher Emotional Labor Strategy Scale adapted by Yin	Teachers’ SA had a significantly positive effect on EE and DP. DA had a significantly negative effect on teachers’ DP and reduced PA.
18	Wróbel (2013)	81.5%	ELE and HS	Poland	MBI	Emotional Labor Scale	DA was a significant mediator in the relationship between empathy and EE
19	Yao et al. (2015)	63.3%	ELE and HS	China	MBI-ES	Teacher Emotional Labor Strategy Scale adapted by Yin	SA positively predicted EE, and DA had no significant effect on EE
20	Yilmaz et al. (2015)	43.6%	ELE and HS	Turkey	MBI	Emotional Labor Scale	SA and GE are the important predictors for both EE and the DP of teachers. PA predicted by EL.
21	Zaretsky et al. (2019)	100%	ELE and HS	Israel	Teacher Burnout Questionnaire	Emotional Labor of Teaching Scale by Levine Brown	High level of TB associates with higher use of SA. Low levels of TB use more GE.

Note. ELE = elementary school, HS = high school, MBI = Maslach Burnout Inventory, EL = emotional labor, SA = surface acting, DA = deep acting, GE = genuine emotions, ΤB = total burnout score, DP = depersonalization, EE = emotional exhaustion, EI = emotional intelligence, PsyCap= psychological capital, OCB—I = organizational citizenship behavior—Interpersonal, PA = personal accomplishment.

## Data Availability

Not applicable.

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
