# Peer review of "Emotional Labor and Burnout among Teachers: A Systematic Review"

_ijerph, 2021, doi:10.3390/ijerph182312760_

Round 1

Reviewer 1 Report

Firstly, I would like to thank the authors for the opportunity of reading and review their interesting manuscript. The paper addresses a topic that is within the journal's scope and very relevant for the present context. The paper basically presents the results of a systematic review. In general terms, the research is well designed and well-conducted, and the results are properly presented. Nevertheless, I have some concerns/suggestions regarding the next manuscript’s sections:

 1. Introduction

- On page 1, on statement 3, I think the authors should indicate some references for supporting their affirmation.

- It would be interesting the author could consult recent research conducted about the application of the Conservation of Resources Model (COR) to the analysis of burnout, as:

Meseguer de Pedro, M., Fernández-Valera, M. M., García-Izquierdo, M., & Soler-Sánchez, M. I. (2021). Burnout, psychological capital and health during COVID-19 social isolation: A longitudinal analysis. International Journal of Environmental Research and Public Health18(3), 1064.

- The section of the Introduction 1.3 should be replaced at first of the Introduction section as a contextualization of the phenomenon.

 MATERIALS AND METHODS

 Search Methods

- It should be specified why the search is between 2006 and 2021. Why do not earlier?

- On future occasions, I advise the authors to register their systematic review in a proper dataset as PROSPERO, for example

- They should specify in a very detailed way, the terms they used for their search in every database

- They should also specify their PICO strategy

- From my point of view, the searching strategy (I mean the terms they used for searching) was not very effective, as they had to filter a lot of papers according to the final studies they finally included. Probably, if the terms they used would have been more specific the number of records identified would have been less. 

- It is important to note how was the process of filtering the papers, what was the role every member of the team had for that purpose?

- I would like to know, how was made the additional search?

- In general, I recommend the authors to read the next paper related to the topic of conducting systematic reviews in the organizational context:

Daniels, K. (2019). Guidance on conducting and reviewing systematic reviews (and meta-analyses) in work and organizational psychology. European Journal of Work and Organizational Psychology28(1), 1-10.

Author Response

We would like to thank reviewer one for their helpful feedback. We have addressed all comments and changed the manuscript accordingly. In the following letter, we will respond to each comment individually. Our responses are in red color.

Reviewer 1

  1. On page 1, on statement 3, I think the authors should indicate some references for supporting their affirmation.

Our response: Thank you for this feedback. We have added references, and the text reads as follows:

A significant amount of emotional labor takes place during teaching (Keller,2014). Teaching is a multitasking profession that consists of both cognitive and emotional components; such as teaching and designing the curriculum but also expressing or hiding true emotions or expressing the appropriate emotion for the situation even if it’s not true (Park, 2014). Teaching can be like acting in a live play or stand-up comedy (Akın,2014). Teachers are constantly exposed to the criticism of their students, parents, coworkers and principals, and are forced to deal with numerous emotional situations and at the same time be a role model for their students (Richardson, 2008; Silbaugh, 2021)

  1. It would be interesting the author could consult recent research conducted about the application of the Conservation of Resources Model (COR) to the analysis of burnout, as: Meseguer de Pedro, M., Fernández-Valera, M. M., García-Izquierdo, M., & Soler-Sánchez, M. I. (2021). Burnout, psychological capital and health during COVID-19 social isolation: A longitudinal analysis. International Journal of Environmental Research and Public Health18(3), 1064.

Our response: We thank the reviewer for this interesting paper. We have included the reflections concerning the COR model from the paper in our introduction section.

  1. The section of the Introduction 1.3 should be replaced at first of the Introduction section as a contextualization of the phenomenon.

Our response: We thank the reviewer for this suggestion, and we have now moved this section to the beginning of the paper.

 MATERIALS AND METHODS

  1. Search Methods: It should be specified why the search is between 2006 and 2021. Why do not earlier?

Our response: We restricted our searches to research published in the previous 15 years 2006–2021. This decision was based on the view that more recent research will use more rigorous methodologies and recent data but will also incorporate important findings from previous research.

  1. On future occasions, I advise the authors to register their systematic review in a proper dataset as PROSPERO, for example.

Our response: We appreciate that we should have registered it with PROSPERO would have been desirable. Unfortunately, when we checked the website we were informed that due to COVID, there would be a long delay (and that this delay would be significantly longer for non-UK applicants). Here is the text of the email:

“There is currently a very high demand for registration, we will aim to respond within 10 working days for UK submissions. For submissions from outside the UK, it will be considerably longer - possibly around three months.”

In the end we decided to proceed with the review and not wait for the registration.

  1. They should specify in a very detailed way, the terms they used for their search in every database.

Our response: We have edited this section to more explicitly describe the search terms.

Initially, ‘emotional labour’ and ‘burnout’ were the search terms used in every database. After excluding duplicates and non-English papers, the following search terms were included in the search process: teacher, schools, emotional exhaustion, elementary school, middle school, high school, primary school, educator.

  1. They should also specify their PICO strategy.

Our response: We have specified that we followed the PRISMA guidelines for systematic reviews, but we did not refer specifically to the use of the PICO strategy in our paper. Therefore, we have edited the text to clearly refer to PICO. The edited text is as follows:

To operationalize the research questions and search strategy, we were guided by the PICOS approach (Shamseer et al., 2015). The PICOS acronym, which stands for population, intervention, comparators, outcomes and study design, was used to refine the research questions and develop the terms used in searching the literature.

The search terms were developed on the basis of the research questions and the inclusion/exclusion criteria detailed earlier. Based on this feedback and the test searches, we made some minor modifications and developed the final search terms for each of the PICOS.

  1. From my point of view, the searching strategy (I mean the terms they used for searching) was not very effective, as they had to filter a lot of papers according to the final studies they finally included. Probably, if the terms they used would have been more specific the number of records identified would have been less. 

Our response: We have explained in more detail our search terms (see answer to point 6). Also, the screening sequence is shown in figure 1. Overall, we didn’t want to miss any papers, so we used broad approach to begin with.

  1. It is important to note how was the process of filtering the papers, what was the role every member of the team had for that purpose?

Our response: Our search strategy began by searching the online databases with the terms emotional labor and burnout for journal articles. We continue by locating the duplicate subscriptions, non-English texts and finally we kept only the full text versions. The filtering process continued by applying the inclusive criteria, teacher profession, school settings, level of school, only quantitative papers

We have described the role of all team members in the section on Author Contributions. We repeat it here:

Author Contributions: AK and AM contributed to the conceptualization of the paper. AK and PK were responsible for searching and identifying the studies to be reviewed. AK, PK, OL and AM also contributed to the writing of the paper. All authors have read and agreed to the published version of the manuscript.

  1. I would like to know, how was made the additional search?

Our response: In the section on search methods (2.2), we refer to the fact in order to account for any studies that did not turn up in the database search, we proceeded to a manual scoping of the cited articles of all eligible studies. This involved checking the references lists of included papers to make sure that no paper has been missed by our search strategy. No new papers were identified using this method.

  1. In general, I recommend the authors to read the next paper related to the topic of conducting systematic reviews in the organizational context:

Daniels, K. (2019). Guidance on conducting and reviewing systematic reviews (and meta-analyses) in work and organizational psychology. European Journal of Work and Organizational Psychology28(1), 1-10.

Our response: We would like to thank reviewer one for this helpful paper. We have addressed all comments and changed the manuscript accordingly.

References

Akın, U., Aydın, İ., Erdoğan, Ç. & Demirkasımoğlu, N. (2014). Emotional labor and burnout among Turkish primary school teachers. The Australian Educational Researcher, 41(2), 155-169.

Keller, M. M., Chang, M. L., Becker, E. S., Goetz, T., & Frenzel, A. C. (2014). Teachers’ emotional experiences and exhaustion as predictors of emotional labor in the classroom: An experience sampling study. Frontiers in psychology, 5, 1442.

Park, H. I., O'Rourke, E., & O'Brien, K. E. (2014). Extending conservation of resources theory: The interaction between emotional labor and interpersonal influence. International Journal of Stress Management, 21(4), 384–405.

Richardson, B. K., Alexander, A., & Castleberry, T. (2008). Examining teacher turnover in low-performing, multi-cultural schools: Relationships among emotional labor, communication symmetry, and intent to leave. Communication Research Reports, 25(1), 10-22.

Shamseer, L., Moher, D., Clarke, M., Ghersi, D., Liberati, A., Petticrew, M., …Stewart, L. A. (2015). Preferred reporting items for systematic review and meta-analysis protocols (PRISMA-P) 2015: Elaboration and explanation. British Medical Journal, 349, g7647

Silbaugh, M. W., Barker, D. B., & Arghode, V. (2021). Emotional Labor, Emotional Intelligence, and Burnout among School Principals: Testing a Mediational Model. Leadership and Policy in Schools, 1-14.

Reviewer 2 Report

Comments and suggestions for authors are included in the attached file

Author Response

We would like to thank reviewer two for their helpful feedback. We have addressed all comments and changed the manuscript accordingly. In the following letter, we will respond to each comment individually.

Reviewer 2

  1. Perhaps the only issues that might need a correction would be the following: Expand the search language to the works published in Spanish

Our response: We restricted searches to English language databases because the research team did not have the capacity to search beyond this. 

  1. Use a term other than "Labor" when talking about "Emotional Labor", since when translating into other languages it is difficult to understand.

Our response: This is interesting point. We are not sure what alternative term that we can use, given that the term ‘Emotional Labor’ is connected with this research field. We would be happy to hear suggestions from the reviewer.

Round 2

Reviewer 1 Report

The authors have arranged all the comments made by the reviewer.